# Factors influencing user decision of telemedicine applications in Thailand

Chadakan Yan[1,2,3], Boonyarat Samphanwattanachai[1], Chitsanupong Ratarat[4], Phichayut Phinyo [2,3,5]*

1 Faculty of Business and Technology, Stamford International University, Bangkok, Thailand, 2 Center for Clinical Epidemiology and Clinical Statistics, Faculty of Medicine, Chiang Mai University, Chiang Mai, Thailand, 3 Department of Biomedical Informatics and Clinical Epidemiology (BioCE), Faculty of Medicine, Chiang Mai University, Chiang Mai, Thailand, 4 Global Business School for Health, Faculty of Population Health Sciences, University College London, London, United Kingdom, 5 Musculoskeletal Science and Translational Research (MSTR) Center, Chiang Mai University, Chiang Mai, Thailand,

* phichayutphinyo@gmail.com

## Abstract

Telemedicine applications have been used worldwide to support the healthcare system in both private and government sectors. However, the factors influencing users' decisions to use the telemedicine application have not been well determined. Exploratory cross-sectional research was conducted using an offline and online questionnaire on Thai individuals aged 18–65. The recruitment period for this study spanned from December 25, 2023, to March 25, 2024, utilizing quota sampling to ensure representation across different regions of Thailand. The objectives were to estimate the proportion of individuals using telemedicine applications and to identify significant determinants of the decision to use telemedicine applications, including Acceptance and Use of Technology, the information systems (IS) Success Model, Trust, and Perceived Risk factors. Exploratory factor analysis (EFA) was used to identify potential latent factors from the 62-item multidimensional questionnaire. Multiple linear regression was used to identify significant determinants of using telemedicine applications. EFA was performed to group 62 variables into 6 latent factors, including trust, ease of use, system quality, benefits of use, price, and service quality. Of 385 Thai individuals, the proportion of those who use telemedicine applications was 63.63%. All six determinants significantly influenced the decision to use telemedicine applications. The factors influencing individuals' decisions to use telemedicine applications include trust, ease of use, system quality, benefits of use, price, and service quality.

## Introduction

Telemedicine delivers healthcare services and consultations where healthcare providers use technologies to diagnose, treat, prevent diseases, and provide health advice to patients remotely [1,2]. This also includes research and evaluation for the

**Data availability statement:** All relevant data are within the manuscript and its Supporting information files.

**Funding:** The author(s) received no specific funding for this work.

**Competing interests:** The authors have declared that no competing interests exist.

**Abbreviations:** EFA, Exploratory Factor Analysis; CFA, Confirmatory Factor Analysis; UTAUT2, The unified Theory of Acceptance and Use of Technology; NHSO: National Health Security Office; COVID-19, Coronavirus Disease 2019; SD, Standard Deviation; PCA, Principal Components Analysis; RATER, Reliability, Assurance, Tangibles, Empathy, Responsiveness

continuous education of medical personnel. Users received telemedicine via various communication channels such as video calls, messaging (chat), and telephone, with data encryption used to secure personal medical information and maintain confidentiality [1,2].

Telemedicine services can be divided into medical information sharing, real-time chat, and video consultation [3,4]. Medical information sharing or store-and-forward telemedicine involves sending patient data, such as history or lab results, to a healthcare provider for consultation [5,6]. Patients and doctors communicate in real-time through text-based messaging, while video consultation enables virtual face-to-face consultations and visual assessments [3,4].

Telemedicine offers many advantages, including increased patient accessibility to healthcare services, reduced overall medical costs, and decreased unnecessary waiting times in hospitals [7]. However, there are challenges, such as potential treatment inaccuracies due to inadequate information, slow connections, and doctors' limited experience with online consultations [8].

The COVID-19 pandemic has led to a rapid adoption of telemedicine services, with a 154% increase observed during this period. In the United States, there was a 50% increase in telemedicine usage in 2020 compared to the same period in 2019 [9]. A large health system in New York reported a sharp increase of up to 683% in daily telemedicine usage [10].

During the COVID-19 pandemic, telemedicine has been widely used in Thailand to increase patient accessibility to healthcare services, including health consultation, diagnosis, and surveillance [11]. Thermometers and pulse oximeters are used to monitor patients at home during the pandemic. If there is an abnormality in oxygen saturation, the patient is notified to receive a teleconsultation with doctors [12]. The National Health Security Office (NHSO) has included telemedicine for reimbursement under the Universal Coverage Scheme, which covers 75% of the population. This service is available for patients with stable chronic diseases such as hypertension, diabetes mellitus, asthma, cancer, and mental illness [13].

There are many telemedicine mobile application platforms, some partnered with NHSO and others independent [14]. However, there is a lack of research on factors influencing users' decisions to use telemedicine applications in Thailand. Therefore, this study aims to determine the factors impacting the decision to use telemedicine applications among Thai people.

## Materials and methods

Exploratory factor research was conducted using a cross-sectional design and a self-reported questionnaire. The objectives of this study were to estimate the prevalence of telemedicine application usage and to identify key determinants influencing the decision to use telemedicine applications among Thai individuals aged 18–65 years. The recruitment period for the study spanned from December 25, 2023, to March 25, 2024, utilizing quota sampling across various regions of Thailand. Written informed consent was obtained at the start of the online survey, explaining the study's purpose, voluntary participation, and that proceeding implied consent. The study was

approved by the Human Research Ethics Committee of Stamford International University (STIU-HREC040/2023). The sample size calculation was based on an infinite population proportion. To define a 95% confidence level and precision of 0.05, a total sample size of 385 was required.

Prior to data collection, a pilot study was conducted with 30 Thai individuals aged 18–65 who had prior experience using telemedicine applications to assess the questionnaire's internal consistency. Cronbach's alpha coefficients for each construct ranged from 0.763 to 0.984, exceeding the accepted threshold of 0.70 [15,16], and indicating good to excellent reliability (S1 Table). This pilot sample was independent of the main study population of 385 participants used for exploratory factor analysis and regression.

A questionnaire was used to collect data from a sample group consisting of 385 Thai citizens, both men and women, aged 18–65 years, who had or had not used telemedicine application services in Thailand. We collected all self-reported demographic data and factors influencing the decision to use telemedicine applications, including demographic data, The unified Theory of Acceptance and Use of Technology (UTAUT2), Information systems (IS) Success Model, Trust, and Perceived Risk using the Five-Point Likert Scale (5=strongly agree, 4=agree, 3=neutral, 2=disagree, 1=strongly disagree). To interpret the average score of the Five-point Likert scale for each factor from the questionnaire, we use class intervals divided into 5 levels with 0.8 intervals in each class, including strongly agree (4.21–5.00), agree (2.61–3.40), neutral (2.61–3.40), disagree (1.81–2.60) and strongly disagree (1.00–1.80). There were 62 questionnaire items from the factors influencing the decision to use telemedicine applications including UTAUT2, IS Success Model, Trust, and Perceived Risk (S1 File).

All statistical analyses were performed using the SPSS program (IBM Corp. Released 2016. IBM SPSS Statistics for Windows, Version 24.0. Armonk, NY: IBM Corp.) The normality of continuous data was examined by visualizing histograms. Continuous data regularly distributed was represented using the mean and standard deviation. Frequency and percentage were employed to characterize categorical data.

Exploratory factor analysis (EFA) was used to study the structure of latent factors and reduce their number by grouping similar factors from all 62 questionnaire items. In conducting EFA, principal components were extracted using Principal Components Analysis (PCA) [17]. Subsequently, orthogonal rotation was applied using the Varimax method to achieve a clearer factor structure. [18] The criteria for determining the appropriate number of factors involve considering eigenvalues greater than 1 and ensuring that the factor loading of each variable is at least 0.5. Additionally, each variable should not have high and similar factor loadings on more than one factor [17]. Multiple linear regression was used to identify significant determinants for using telemedicine applications, with a P-value of less than 0.05 considered statistically significant.

## Results

Of the 385 Thai respondents (95.77% response rate; 4.22% incomplete responses), 63.63% (245/385) reported having used telemedicine applications. The sociodemographic characteristics of the respondents and their telemedicine usage behaviors are presented in Tables 1 and 2, respectively.

EFA was performed (S2–S4 Tables) to group the 62 questionnaire items (From Acceptance and Use of Technology, IS Success Model, Trust, and Perceived Risk factors questionnaires) into 6 relevant factors; Trust, Ease of Use, System Quality, Benefits of Use, Price, and Service Quality. The 6 factors represented 76.48% (R-squared) of all variance (S3 Table).

From multiple linear regression analysis of 6 factors, all factors were statistically significant with an R-squared of 0.670 (Table 3). The Beta Coefficient ranked as follows: Trust (Beta=0.490), System Quality (Beta=0.456), Ease of Use (Beta=0.274), Cost (Beta=0.271), Benefit of Use (Beta=0.243), and Service Quality (Beta=0.121). The equation below depicts the decision-making process for using telemedicine applications based on six significant independent variables. Each factor's impact is reflected by its beta value (Fig 1).

$$Y = +3.401 + 0.531(X_1) + 0.297(X_2) + 0.494(X_3) + 0.263(X_4) + 0.293(X_5) + 0.131(X_6) \tag{1}$$

**Table 1. Sociodemographic characteristics of the respondents.**

| Characteristics | Number (n) | Percentage |
|---|---|---|
| **Age** | | |
| 18–30 | 69 | 17.92 |
| 31–40 | 147 | 38.18 |
| 41–50 | 104 | 27.01 |
| 51 and above | 65 | 16.88 |
| **Sex** | | |
| Male | 259 | 67.27 |
| Female | 126 | 32.73 |
| **Education level** | | |
| Below bachelor's degree | 37 | 9.61 |
| Bachelor's degree | 253 | 65.71 |
| Postgraduate Degree | 95 | 24.68 |
| **Monthly Income (Thai Baht)** | | |
| < 10,000 | 38 | 9.87 |
| 10,000–20,000 | 177 | 45.97 |
| 20,001–30,000 | 128 | 33.25 |
| > 30,000 | 42 | 10.91 |
| **Region** | | |
| Bangkok | 32 | 8.31 |
| Central (Excluding Bangkok) | 73 | 18.96 |
| North | 69 | 17.92 |
| Northeast | 129 | 33.51 |
| South | 53 | 13.77 |
| East | 29 | 7.53 |
| **Medical Condition** | | |
| Without any underlying disease | 303 | 78.70 |
| With underlying diseases | 82 | 21.30 |

Where  $X_1$ = Trust
$X_2$ = Ease of Use
$X_3$ = Cost
$X_4$ = Benefit of Use
$X_5$ = Cost
$X_6$ = Service Quality
$Y$ = Decision to use telemedicine application

## Discussion

This cross-sectional study utilized a self-administered questionnaire to identify factors associated with individuals' decisions to use telemedicine applications. Multiple linear regression analysis identified six key factors influencing individuals' decisions to use telemedicine applications: trust, cost, ease of use, system quality, perceived benefits, and service quality.

### Trust

Trust is the most important factor for choosing telemedicine applications in our study. It means that the platform is reliable, and users believe it can cater to their health needs while protecting their privacy and confidentiality [19]. Trust in the

**Table 2. Behavior of using a telemedicine application.**

| Frequency of Use of Telemedicine Application Services | Number (n) | Percentage |
|---|---|---|
| Once a month | 13 | 3.38 |
| Once a year | 193 | 50.13 |
| Two to five times a year | 91 | 23.64 |
| Once every few years | 88 | 22.86 |
| **Average Cost of Telemedicine Application Service Per Visit (Thai Baht)** | **Number (n)** | **Percentage** |
| Less than 500 | 69 | 17.92 |
| 500 − 1,000 | 199 | 51.69 |
| 1,001 - 2,000 | 105 | 27.27 |
| 2,001 - 3,000 | 12 | 3.12 |
| **Reasons For Using Telemedicine Application Services** | **Number (n)** | **Percentage** |
| Health Promotion | 77 | 20.00 |
| Seeking Treatments | 241 | 62.60 |
| Exercise | 20 | 5.19 |
| Leisure | 18 | 4.68 |
| Health Check-up | 29 | 7.53 |
| **The Person Involved in Decision-Making for Using Telemedicine Application Services.** | **Number (n)** | **Percentage** |
| Themselves | 211 | 54.81 |
| Friends | 94 | 24.42 |
| Family Members | 29 | 7.53 |
| Supervisor/Organization/Company | 31 | 8.05 |
| Salesperson | 20 | 5.19 |
| **Source for information about Telemedicine Applications** | **Number (n)** | **Percentage** |
| Print Media (Books, Magazine, Newspapers) | 18 | 4.68 |
| Social Media | 247 | 64.16 |
| Websites | 120 | 31.17 |

**Table 3. Multiple linear regression analysis of factors influencing telemedicine application usage decisions.**

| Model | | Unstandardized Coefficients | | Standardized Coefficients | t | P-values |
|---|---|---|---|---|---|---|
| | | Beta | Standard error | Beta | | |
| 1 | (Constant) | 3.401 | 0.032 | | 106.543 | <0.001 |
| | Trust | 0.531 | 0.032 | 0.490 | 16.607 | <0.001 |
| | Ease of Use | 0.297 | 0.032 | 0.274 | 9.297 | <0.001 |
| | System Quality | 0.494 | 0.032 | 0.456 | 15.458 | <0.001 |
| | Benefit of Use | 0.263 | 0.032 | 0.243 | 8.242 | <0.001 |
| | Cost | 0.293 | 0.032 | 0.271 | 9.175 | <0.001 |
| | Service Quality | 0.131 | 0.032 | 0.121 | 4.106 | <0.001 |

$R = 0.810$, $R^2 = 0.670$, Adj $R^2 = 0.665$, Standard error = 0.625, Sig of F = 0.000.

healthcare provider is also vital to patients' decisions, as it demonstrates that their physician can effectively treat them through the telemedicine application. This aligns with findings from other digital platforms in different industries, where providers' success relies heavily on users' trust [20,21]. Similarly, previously published studies indicate that trust significantly influences the decision to choose telemedicine services. However, it's not only trust in the platform that matters but

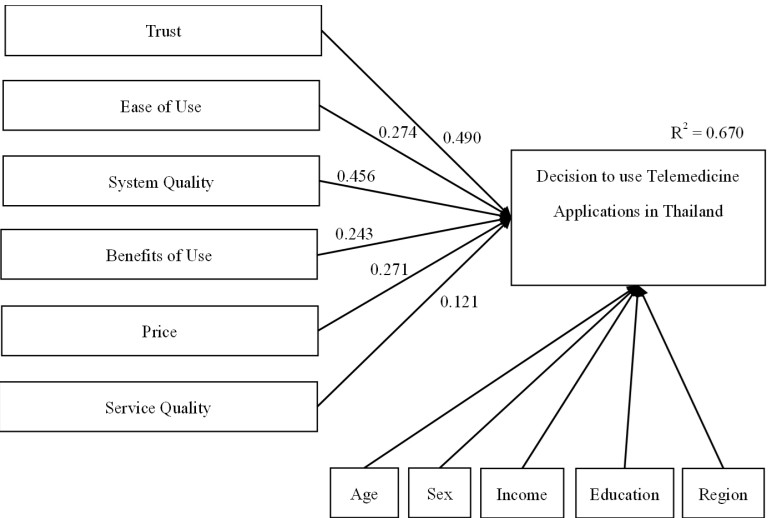

**Fig 1. Factors Influencing User Decision of Telemedicine Applications in Thailand.**

also trust in the information provided by the physician. Users need to feel confident that the advice and diagnoses they receive are accurate and trustworthy [22].

## Ease of use

Ease of use means the application is intuitive, easy to navigate, and compatible with multiple devices. Perceived ease of use positively impacts users' acceptance of mobile health applications, reducing the effort required to learn and operate the application. Similarly, previous research combining the Technology Acceptance Model and the IS Success Model to analyze e-health utilization found that ease of use significantly influences the intention to use e-health services [23]. This finding aligns with a study conducted in China, which examined the intention to use telemedicine services during the COVID-19 pandemic and found that perceived ease of use is critical in encouraging continued usage [24]. When applications are user-friendly, they not only attract initial users but also promote ongoing engagement and satisfaction [25].

## System quality

System quality refers to the telemedicine application's stability, notification system, responsiveness, data transmission speed, and accuracy. The applications need to track patients' progression, exchange information, and solve patients' problems. This finding supports results from previously published studies on telemedicine usage during the pandemic in Nigeria [26] and Pakistan [27], as well as the decision of health professionals in Yemen to use telemedicine services [28].

## Benefit of use

The benefits to the user extend beyond improved health outcomes to include enhanced health knowledge, which facilitates a better understanding of personal health issues, and significant time savings from reduced travel to healthcare facilities. Users who perceive the application as beneficial and effective for managing their health conditions are more likely to engage with mobile health applications [29]. The perception of benefits and user engagement is evident not only in telemedicine applications but also in other mobile applications, such as social media [30]. This phenomenon is supported by research on the usage of e-health and telemedicine services during the COVID-19 pandemic [23,31].

## Cost

Cost refers to the expense of using the telemedicine application, including free subscription options and the overall cost-effectiveness of the service. Price is an important factor that patients use to weigh the perceived benefits of the telemedicine application against its financial cost. The lower the cost of using the technology, the higher the adoption rate [32]. This is because affordable pricing makes the service accessible to a broader range of users, encouraging initial adoption. Moreover, cost is influential in the decision to use telemedicine applications and plays a significant role in the decision to continue using them. Users are more likely to remain loyal to a telemedicine service if they find it cost-effective, perceiving that the benefits they receive justify the price they pay. Additionally, users who continue to use the service are more inclined to recommend telemedicine applications to others [27,33].

## Service quality

Healthcare providers' responsiveness and willingness to provide service, including hospitality, and understanding of patients, can lead to a good rapport and the perception of high service quality. High quality service results in customer satisfaction and loyalty [34]. This is also evident in the acceptance of new technology in healthcare. A study in Indonesia on the success factors of implementing electronic medical records found that service quality significantly affects the success of implementation [35]. Similar findings are reported in other research on the quality of service and the use of telemedicine systems.

Moreover, sociodemographic factors, including age, gender, education, income, and region were found significantly different among the respondents which might considered as one of the influencing factors. The age group 41−50 uses telemedicine applications the most, followed by 31–40-year-olds. This contrasts with other studies indicating that people under 35 use mobile health applications more frequently [24,36–39]. Men are more likely to use telemedicine applications than women in Thailand, which differs from some studies suggesting women use these services more [40,41]. Men typically prefer fitness applications, while women favor nutrition, self-care, and reproductive health apps [24]. Individuals with postgraduate education are more likely to use telemedicine services than those with only a bachelor's degree in Thailand. Higher education levels correlate with better technology understanding, health literacy, and financial means to afford telemedicine services [42,43]. People with incomes higher than 30,000 THB are the most frequent users of telemedicine services, consistent with studies showing that higher-income families tend to use more telemedicine services [42,44,45]. Telemedicine application usage is higher in Bangkok than in other regions. Urban areas show a greater increase in telemedicine usage compared to rural areas [46], especially noted during the COVID-19 pandemic [47].

This research examined factors influencing patients' decisions to use telemedicine applications in Thailand. The evaluation was based on Trust, Cost, Ease of use, System Quality, Benefits of Use, and Service Quality. A factor analysis assesses these impacts, achieving an R-squared value of 0.670, indicating that the model explains 67% of the variance, with the remaining 33% attributed to other factors.

The findings of this study are consistent with international evidence on telemedicine adoption across varied healthcare contexts. Prior research from high- and middle-income countries, including the United States, China, and several European nations, has consistently identified trust, perceived ease of use, system quality, and cost as key determinants influencing the uptake of telemedicine services [48–50]. For instance, a systematic review by Almathami et al. emphasized that trust in both healthcare providers and digital platforms is critical for the acceptance and sustained use of real-time, home-based telemedicine consultations [48]. In the context of China, Guo et al. demonstrated that perceived ease of use and service quality significantly predicted users' intention to adopt mobile health services, particularly under heightened demand during the COVID-19 pandemic [49]. Moreover, a global systematic review by Kruse et al. underscored the role of cost as a significant barrier to telemedicine implementation, especially in low- and middle-income settings, where affordability can critically affect both initial adoption and continued use [50]. The concordance between these international

findings and the present study reinforces the generalizability of the identified determinants and highlights the necessity of user-centered, context-specific strategies to support the sustainable adoption of telemedicine services.

This study is subject to several limitations. First, its cross-sectional design precludes the establishment of causal relationships between the identified factors and telemedicine application usage. Second, the reliance on self-reported data introduces potential recall and response bias. Third, the use of quota sampling may limit the generalizability of findings to the broader Thai population, particularly among individuals without internet access. Moreover, the study included only individuals aged 18–65 years with prior telemedicine experience, thereby excluding younger and older age groups who may exhibit different usage patterns. Lastly, although EFA was utilized to identify latent constructs, confirmatory factor analysis (CFA) is warranted in future studies to validate the factor structure and ensure construct validity.

Future research should aim to strengthen the external validity and contextual understanding of telemedicine adoption. Qualitative studies involving in-depth interviews with patients and healthcare professionals could yield nuanced insights into the facilitators and barriers to utilization. Additional factors such as clinical outcomes, health literacy, and self-efficacy should be considered, given their potential influence on technology engagement. Stratification of services by disease group may also be beneficial, as certain conditions necessitating physical examination may be less amenable to virtual care modalities. Furthermore, special attention should be directed toward vulnerable populations, including the elderly and pediatric patients, to promote equitable access and minimize disparities in digital health adoption.

From a service delivery perspective, private-sector telemedicine providers may benefit from integrating business frameworks to enhance quality and patient engagement. The marketing mix framework can inform targeted improvements in areas such as pricing strategy, service accessibility, and promotional communication. In parallel, the RATER model—comprising Reliability, Assurance, Tangibles, Empathy, and Responsiveness—offers a validated structure for assessing and improving service quality. Leveraging such models may help identify areas requiring optimization, thereby improving patient satisfaction, fostering loyalty, and contributing to the long-term sustainability of telemedicine platforms.

## Conclusion

The decision to use telemedicine applications in Thailand is influenced by several critical factors. Trust in the platform and healthcare providers, ease of use, system quality, perceived benefits, cost, and service quality significantly impact user adoption. Demographic factors such as age, gender, income, education level, and geographical location also play an important role. These findings suggest that enhancing trust, maintaining high system quality and service, and addressing demographic-specific needs are necessary to achieve the adoption and sustained use of telemedicine applications in Thailand.

## Supporting information

**S1 Table. Reliability Test of Survey Data.**
(DOCX)

**S1 File. The questionnaire in the English version.**
(PDF)

**S2 Table. Kaiser-Meyer-Olkin Measure and Bartlett's Test of 62 variables.**
(DOCX)

**S3 Table. Principal Component Analysis (PCA).**
(DOCX)

**S4 Table. Factor loading using Varimax rotation.**
(DOCX)

## Acknowledgments

This study was partially supported by the Faculty of Medicine, Chiang Mai University and Chiang Mai University.

## Author contributions

**Conceptualization:** Chadakan Yan, Boonyarat Samphanwattanachai, Phichayut Phinyo.

**Data curation:** Chadakan Yan, Boonyarat Samphanwattanachai, Chitsanupong Ratarat, Phichayut Phinyo.

**Formal analysis:** Chadakan Yan, Boonyarat Samphanwattanachai, Chitsanupong Ratarat, Phichayut Phinyo.

**Investigation:** Chadakan Yan, Boonyarat Samphanwattanachai, Chitsanupong Ratarat, Phichayut Phinyo.

**Methodology:** Chadakan Yan, Boonyarat Samphanwattanachai, Chitsanupong Ratarat, Phichayut Phinyo.

**Project administration:** Boonyarat Samphanwattanachai, Phichayut Phinyo.

**Resources:** Phichayut Phinyo.

**Software:** Phichayut Phinyo.

**Writing – original draft:** Chadakan Yan.

**Writing – review & editing:** Boonyarat Samphanwattanachai, Chitsanupong Ratarat, Phichayut Phinyo.

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
