## [Decision Letter · Decision Letter 0]

16 Apr 2025

PONE-D-25-05830Factors Influencing User Decision of Telemedicine Applications in ThailandPLOS ONE

Dear Dr. Phinyo,

Thank you for submitting your manuscript to PLOS ONE. After careful consideration, we feel that it has merit but does not fully meet PLOS ONE’s publication criteria as it currently stands. Therefore, we invite you to submit a revised version of the manuscript that addresses the points raised during the review process.

Kindly review the manuscript with respect to sample size for the study and the sample used for reliability of the questionnaire. Please mention the limitations of the study. 

We look forward to receiving your revised manuscript.

Kind regards,

Alexander Maniangat Luke, PhD

Academic Editor

PLOS ONE

Additional Editor Comments:

The author needs to ensure that the sample size for the piloting is clearly mentioned as opposed to the total sample used for analysis. The mention of the study limitations will enhance the quality of the paper

Reviewers' comments:

Reviewer's Responses to Questions

**Comments to the Author**

1. Is the manuscript technically sound, and do the data support the conclusions?

Reviewer #1: Yes

Reviewer #2: Yes

Reviewer #3: Yes

2. Has the statistical analysis been performed appropriately and rigorously? 

Reviewer #1: Yes

Reviewer #2: I Don't Know

Reviewer #3: Yes

3. Have the authors made all data underlying the findings in their manuscript fully available?

Reviewer #1: Yes

Reviewer #2: Yes

Reviewer #3: Yes

4. Is the manuscript presented in an intelligible fashion and written in standard English?

Reviewer #1: Yes

Reviewer #2: Yes

Reviewer #3: Yes

5. Review Comments to the Author

Reviewer #1: Interesting study.

1) Abstract = okay

2)Introduction = okay

3) Materials and methods: I would like to understand why the statistic for individuals who have been using the telemedicine application was only 30, while for individuals who have used telemedicine application services in Thailand it was 385. The discrepancy between the numbers seems quite large to me.

4) Results= okay

5) Discussion = Okay

6) Conclusion+ Okay

Reviewer #2: This manuscript presents a well-structured and relevant study on the factors influencing the adoption of telemedicine applications in Thailand. Here are some suggestions for the authors:

The methodology is sound, with appropriate use of exploratory factor analysis and regression, and the findings—highlighting trust, ease of use, system quality, benefits, price, and service quality—are both timely and impactful.

The writing is generally clear, though a few typographical errors (e.g., “Decisio” in keywords and “service quality” in the abstract) should be corrected.

The study would benefit from a clearer summary of the factor analysis results, perhaps in the form of a table or diagram. Additionally, briefly situating the findings within global telemedicine adoption literature could strengthen the discussion.

Reviewer #3: The study has been well presented with apropriate sections. The methodology part has been well described and is robust, but nevertheless a seperate paragraph on limitations will improve the readablity of your study

6. PLOS authors have the option to publish the peer review history of their article (what does this mean? ). If published, this will include your full peer review and any attached files.

**Do you want your identity to be public for this peer review?** For information about this choice, including consent withdrawal, please see our Privacy Policy .

Reviewer #1: No

Reviewer #2: No

Reviewer #3: **Yes: ** sudhir Rama varma

---

## [Author Response · Author response to Decision Letter 1]

17 Apr 2025

Response to Reviewers

We sincerely thank the editor and reviewers for their thoughtful comments and constructive feedback, which have significantly strengthened the manuscript. We have addressed each comment below in detail and indicated specific line numbers in the revised manuscript where changes were made. All edits were made in accordance with PLOS ONE’s formatting and editorial requirements.

Manuscript Title: Factors Influencing User Decision of Telemedicine Applications in Thailand

Manuscript ID: PONE-D-25-05830

Editor Comments

Response: We have revised the manuscript to conform to PLOS ONE formatting guidelines, including file naming and structure, as recommended.

Response: The ethics statement now appears exclusively in the Materials and Methods section (Lines 85-86). It has been removed from all other sections.

Response: The reference list has been carefully reviewed for completeness and correctness. No retracted articles are cited. All references have been updated and formatted according to PLOS ONE guidelines.

Additional Editor Comments:

The author needs to ensure that the sample size for the piloting is clearly mentioned as opposed to the total sample used for analysis. The mention of the study limitations will enhance the quality of the paper

Response:

Clarification of Sample Size for Pilot Study

We have clarified the distinction between the pilot sample (n = 30) and the main study sample (n = 385) in the Materials and Methods section. The revised text (Lines 89-94) reads:

“Prior to data collection, a pilot study was conducted with 30 Thai individuals aged 18–65 who had prior experience using telemedicine applications to assess the questionnaire’s internal consistency. Cronbach’s alpha coefficients for each construct ranged from 0.763 to 0.984, exceeding the accepted threshold of 0.70 [15, 16], and indicating good to excellent reliability (S1 Table). This pilot sample was independent of the main study population of 385 participants used for exploratory factor analysis and regression.”

Limitations Section

We have added a clearly defined limitations paragraph to the Discussion section (Lines 239-247), as suggested:

“This study is subject to several limitations. First, its cross-sectional design precludes the establishment of causal relationships between the identified factors and telemedicine application usage. Second, the reliance on self-reported data introduces potential recall and response bias. Third, the use of quota sampling may limit the generalizability of findings to the broader Thai population, particularly among individuals without internet access. Moreover, the study included only individuals aged 18–65 years with prior telemedicine experience, thereby excluding younger and older age groups who may exhibit different usage patterns. Lastly, although exploratory factor analysis (EFA) was utilized to identify latent constructs, confirmatory factor analysis (CFA) is warranted in future studies to validate the factor structure and ensure construct validity.”

Reviewers' comments

Reviewer's Responses to Questions

Comments to the Author

1. Is the manuscript technically sound, and do the data support the conclusions?

Reviewer #1: Yes

Reviewer #2: Yes

Reviewer #3: Yes

Response: None

2. Has the statistical analysis been performed appropriately and rigorously?

Reviewer #1: Yes

Reviewer #2: I Don't Know

Reviewer #3: Yes

Response: None

3. Have the authors made all data underlying the findings in their manuscript fully available?

Reviewer #1: Yes

Reviewer #2: Yes

Reviewer #3: Yes

Response: None

4. Is the manuscript presented in an intelligible fashion and written in standard English?

Reviewer #1: Yes

Reviewer #2: Yes

Reviewer #3: Yes

Response: None

5. Review Comments to the Author

Reviewer #1: Interesting study.

1) Abstract = okay

2) Introduction = okay

3) Materials and methods: I would like to understand why the statistic for individuals who have been using the telemedicine application was only 30, while for individuals who have used telemedicine application services in Thailand it was 385. The discrepancy between the numbers seems quite large to me.

4) Results= okay

5) Discussion = Okay

6) Conclusion+ Okay

Response: We appreciate the reviewer’s observation and welcome the opportunity to clarify this point. The two sample sizes served distinct methodological purposes within the study design.

The 30 participants were recruited solely for a pilot study to assess the internal consistency of the self-developed questionnaire. This step was conducted prior to the main data collection phase. The pilot sample consisted of Thai individuals aged 18–65 who had prior experience using telemedicine applications. Their responses were used to calculate Cronbach’s alpha coefficients for each domain in the instrument. All values exceeded the recommended threshold of 0.70, indicating satisfactory internal consistency. Importantly, this pilot sample was independent of the primary study sample used in the final analysis.

The use of 30 participants in pilot reliability testing is supported in the literature. According to methodological guidelines, a sample size of 30 is generally considered sufficient to evaluate reliability measures such as Cronbach’s alpha in early-stage instrument validation, particularly when the aim is to refine survey tools rather than perform inferential analysis [1-3].

References:

1. Hertzog MA. Considerations in determining sample size for pilot studies. Res Nurs Health. 2008;31(2):180–191. doi:10.1002/nur.20247

2. Johanson GA, Brooks GP. Initial scale development: Sample size for pilot studies. Educ Psychol Meas. 2010;70(3):394–400. doi:10.1177/0013164409355692

3. Van Teijlingen ER, Hundley V. The importance of pilot studies. Soc Res Update. 2001;35(1):1–4.

The 385 participants included in the main study were recruited through quota sampling across different regions of Thailand. This sample was used for exploratory factor analysis and multiple linear regression to identify determinants of telemedicine application usage. The sample size was calculated to ensure sufficient statistical power and generalizability of the findings.

We have revised the Methods section (Lines 89-94) to make this distinction clearer in the manuscript.

“Prior to data collection, a pilot study was conducted with 30 Thai individuals aged 18–65 who had prior experience using telemedicine applications to assess the questionnaire’s internal consistency. Cronbach’s alpha coefficients for each construct ranged from 0.763 to 0.984, exceeding the accepted threshold of 0.70 [15, 16], and indicating good to excellent reliability (S1 Table). This pilot sample was independent of the main study population of 385 participants used for exploratory factor analysis and regression.”

Reviewer #2: This manuscript presents a well-structured and relevant study on the factors influencing the adoption of telemedicine applications in Thailand. Here are some suggestions for the authors:

The methodology is sound, with appropriate use of exploratory factor analysis and regression, and the findings—highlighting trust, ease of use, system quality, benefits, price, and service quality—are both timely and impactful.

The writing is generally clear, though a few typographical errors (e.g., “Decisio” in keywords and “service quality” in the abstract) should be corrected.

The study would benefit from a clearer summary of the factor analysis results, perhaps in the form of a table or diagram. Additionally, briefly situating the findings within global telemedicine adoption literature could strengthen the discussion.

Response: Thank you for the suggestion. All typographical errors in the abstract and keywords have been corrected. We have included summary tables (S2–S4) in the Supporting Information to report the results of the factor analysis, including factor loadings and explained variance. These results are described in the Results section (Lines 129-132). We have added a paragraph to the Discussion section (Lines 225-238) to contextualize our findings within the global literature. The paragraph reads: “The findings of this study are consistent with international evidence on telemedicine adoption across varied healthcare contexts. Prior research from high- and middle-income countries, including the United States, China, and several European nations, has consistently identified trust, perceived ease of use, system quality, and cost as key determinants influencing the uptake of telemedicine services [48-50]. For instance, a systematic review by Almathami et al. emphasized that trust in both healthcare providers and digital platforms is critical for the acceptance and sustained use of real-time, home-based telemedicine consultations [48]. In the context of China, Guo et al. demonstrated that perceived ease of use and service quality significantly predicted users’ intention to adopt mobile health services, particularly under heightened demand during the COVID-19 pandemic [49]. Moreover, a global systematic review by Kruse et al. underscored the role of cost as a significant barrier to telemedicine implementation, especially in low- and middle-income settings, where affordability can critically affect both initial adoption and continued use [50]. The concordance between these international findings and the present study reinforces the generalizability of the identified determinants and highlights the necessity of user-centered, context-specific strategies to support the sustainable adoption of telemedicine services.”

Reviewer #3: The study has been well presented with appropriate sections. The methodology part has been well described and is robust, but nevertheless a separate paragraph on limitations will improve the readability of your study

Response: We agree and have now included a standalone limitations paragraph in the Discussion section (Lines 239-247), as detailed above.

6. PLOS authors have the option to publish the peer review history of their article (what does this mean?). If published, this will include your full peer review and any attached files.

Do you want your identity to be public for this peer review? For information about this choice, including consent withdrawal, please see our Privacy Policy.

Reviewer #1: No

Reviewer #2: No

Reviewer #3: Yes: sudhir Rama varma

Response: None

We are grateful for the opportunity to revise our manuscript. We believe the revisions have improved the manuscript and fully addressed the comments raised by the reviewers and editorial team. Thank you for your time and consideration.

Sincerely,

Phichayut Phinyo, M.D., M.Sc., Ph.D.

Email: phichayutphinyo@gmail.com

---

## [Decision Letter · Decision Letter 1]

15 May 2025

Factors Influencing User Decision of Telemedicine Applications in Thailand

PONE-D-25-05830R1

Dear Dr. Phinyo,

We’re pleased to inform you that your manuscript has been judged scientifically suitable for publication and will be formally accepted for publication once it meets all outstanding technical requirements.

Kind regards,

Alexander Maniangat Luke, PhD

Academic Editor

PLOS ONE

Additional Editor Comments (optional):

Reviewers' comments:

Reviewer's Responses to Questions

**Comments to the Author**

1. If the authors have adequately addressed your comments raised in a previous round of review and you feel that this manuscript is now acceptable for publication, you may indicate that here to bypass the “Comments to the Author” section, enter your conflict of interest statement in the “Confidential to Editor” section, and submit your "Accept" recommendation.

Reviewer #1: All comments have been addressed

Reviewer #3: All comments have been addressed

2. Is the manuscript technically sound, and do the data support the conclusions?

Reviewer #1: Yes

Reviewer #3: Yes

3. Has the statistical analysis been performed appropriately and rigorously? 

Reviewer #1: I Don't Know

Reviewer #3: Yes

4. Have the authors made all data underlying the findings in their manuscript fully available?

Reviewer #1: Yes

Reviewer #3: Yes

5. Is the manuscript presented in an intelligible fashion and written in standard English?

Reviewer #1: Yes

Reviewer #3: Yes

6. Review Comments to the Author

Reviewer #1: The revisions suggested by this reviewer have been made. I have only one comment: I believe the Abstract should include more information about the results. In its current form, only the last sentence provides the information that truly reflects what we expect to learn from the article.

Reviewer #3: The suggestions and remarks highlighted have been addressed well. Appropriate rebuttal has been provided.

7. PLOS authors have the option to publish the peer review history of their article (what does this mean? ). If published, this will include your full peer review and any attached files.

**Do you want your identity to be public for this peer review?** For information about this choice, including consent withdrawal, please see our Privacy Policy .

Reviewer #1: No

Reviewer #3: **Yes: ** Sudhir Rama Varma

---

## [Editor Report · Acceptance letter]

PONE-D-25-05830R1

PLOS ONE

Dear Dr. Phinyo,

I'm pleased to inform you that your manuscript has been deemed suitable for publication in PLOS ONE. Congratulations! Your manuscript is now being handed over to our production team.

Kind regards,

on behalf of

Dr. Alexander Maniangat Luke

Academic Editor

PLOS ONE